# Mucosal Immunization Has Benefits over Traditional Subcutaneous Immunization with Group A Streptococcus Antigens in a Pilot Study in a Mouse Model

**DOI:** 10.3390/vaccines11111724

**Published:** 2023-11-17

**Authors:** Helen Alexandra Shaw, Alex Remmington, Giselle McKenzie, Caroline Winkel, Fatme Mawas

**Affiliations:** Vaccines Division, Science, Research & Innovation, Medicines and Healthcare Products Regulatory Agency, Potters Bar EN6 3QG, UK

**Keywords:** group A Streptococcus, strep A, *Streptococcus pyogenes*, vaccines, multicomponent, mucosal, sublingual, intranasal

## Abstract

Group A Streptococcus (GAS) is a major human pathogen for which there is no licensed vaccine. To protect against infection, a strong systemic and mucosal immune response is likely to be necessary to prevent initial colonization and any events that might lead to invasive disease. A broad immune response will be necessary to target the varied GAS serotypes and disease presentations. To this end, we designed a representative panel of recombinant proteins to cover the stages of GAS infection and investigated whether mucosal and systemic immunity could be stimulated by these protein antigens. We immunized mice sublingually, intranasally and subcutaneously, then measured IgG and IgA antibody levels and functional activity through in vitro assays. Our results show that both sublingual and intranasal immunization in the presence of adjuvant induced both systemic IgG and mucosal IgA. Meanwhile, subcutaneous immunization generated only a serum IgG response. The antibodies mediated binding and killing of GAS cells and blocked binding of GAS to HaCaT cells, particularly following intranasal and subcutaneous immunizations. Further, antigen-specific assays revealed that immune sera inhibited cleavage of IL-8 by SpyCEP and IgG by Mac/IdeS. These results demonstrate that mucosal immunization can induce effective systemic and mucosal antibody responses. This finding warrants further investigation and optimization of humoral and cellular responses as a viable alternative to subcutaneous immunization for urgently needed GAS vaccines.

## 1. Introduction

Group A Streptococcus (GAS, Strep A, *Streptococcus pyogenes*) is a human pathogen that imposes an enormous global burden. It is estimated that over 500,000 deaths annually are attributable to GAS infection, with a significant effect on disability-adjusted life years (DALYs), particularly in low- and middle-income countries (LMICs) [1]. The burden is highest in LMICs where genetic predisposition and poor access to both screening and antibiotic treatments are commonly cited as contributing factors [2,3]. For example, in New Zealand, those of Māori and Pacific Island descent have increased risk of rheumatic heart disease (RHD), with independent contributions to this risk from ethnicity, socioeconomic deprivation and their geographic location [4]. This disparity, as well as the significant impact of post-infection autoimmune sequelae such as RHD, have led the WHO to highlight GAS as a disease of interest for accelerated vaccine development [5]. A roadmap has been established in which prevention of mucosal and skin infections such as pharyngitis and impetigo are the desired endpoint of vaccine candidates as a precursor to prevention of RHD [6]. As the disease burden is high in LMICs, it is important that any future vaccine be affordable, easy to administer and low-risk to ensure global vaccine rollout and uptake.

GAS is a complex organism with a large range of disease manifestations. Infections can range from mild symptoms (pharyngitis and impetigo) to more complicated systemic infections (toxic shock syndrome and necrotizing fasciitis), as well as meningitis in rare cases and autoimmune sequelae such as RHD following repeated exposure. With the initial WHO endpoint targeting pharyngitis, it would be logical to target antigens important in colonization of the oral cavity; however, those involved in invasion and systemic infection should not be discounted because targeting them may help to prevent escape infections. The use of combination vaccines might thus be advantageous to ensure that multiple stages of GAS pathogenesis are covered, with the rationale being that protection is thereby conferred against both milder and more severe manifestations. The vaccines that have made the most progress towards clinical trials and those already in clinical trials either target the M-protein or are multi-component [7]. M-protein vaccines cover either multiple serotypes (30-valent StreptAnova) or the conserved domain (J8, P*17, StrepInCor), while multi-component candidates have mainly focused on conserved antigens such as the Group A Carbohydrate, ScpA, SpyCEP, SpyAD and SLO (Combo 4, Combo 5, VaxA1), as well as multivalent formulations of pilin protein T-antigen (TeeVax) or a combination of J8/p*17 with a minimal epitope peptide from SpyCEP [7,8,9].

An immune response at mucosal sites will be necessary to target these initial colonization and infection events. Studies have indicated that immunization at mucosal sites can lead to an effective immune response to bacterial antigens for several pathogens [10,11,12]. In a passive-protection study, mice were shown to be more significantly protected from intranasal challenge with GAS when it was co-administered with human salivary IgA than when it was co-administered with serum IgG or no treatment [13]. This result was an early indication that mucosal immunity is important to preventing GAS infection. Other studies have shown that an IgA response can be generated with lipoteichoic acid [14], pilin proteins [15] and M protein [16], as well as through the use of virus-like particles (VLPs) [17]. A study in mouse models using a lipopeptide construct containing the J14 minimal M-protein peptide showed that significant IgG and IgA responses were generated following intranasal immunization, such that death and colonization were reduced in treated mice compared with mice given non-adjuvanted and PBS controls [18]. There has been minimal study of combination vaccines using mucosal immunization for induction of mucosal immunity and protection, with most studies focusing on intramuscular and subcutaneous immunization to prevent systemic challenge. Recent studies have started to shift this focus, and a study of the combination vaccine p*17 and K4S2 minimal epitope peptides demonstrated that upper respiratory tract and systemic infection following skin colonization could be almost eradicated through intramuscular immunization followed by intranasal immunization [8]. Another study has investigated sublingual immunization with promising IgA titers against M-protein on VLPs [17].

In this study, we investigated mucosal immunization in comparison to classic subcutaneous immunization using a multi-component vaccine containing GAS recombinant antigens SpyCEP, Cpa, Mac and MalE, as well as a potent mucosal adjuvant (the non-toxic B subunit of cholera toxin). The purpose was to compare the magnitude of the systemic and mucosal antibody responses induced by subcutaneous and mucosal (intranasal and sublingual) immunization and to assess the in vitro success of the responses in impeding adhesion to human epithelial cells, neutralizing virulence-factor activity, and mediating opsonophagocytotic killing.

## 2. Materials and Methods

### 2.1. Bacterial Growth Conditions

*E. coli* were routinely grown in LB and LB agar supplemented with 50 μg/mL kanamycin for selection of pET28a plasmids [19]. Overnight express Instant TB medium supplemented with kanamycin was used to grow cultures for protein expression and purification. NCTC8198 M1 GAS were routinely grown in THY broth (Todd Hewitt supplemented with 0.2% yeast extract) and plated on TH agar (Todd Hewitt supplemented with 1% agar). Frozen cultures for opsonophagocytosis assays (OPA) and binding assays were grown in THY to OD 0.6–0.8 and diluted 1:1 with TH and then 2.5:1 with 70% glycerol before they were frozen at −80 °C in 0.5 mL aliquots. For binding assays, cultures were grown in THY with 30 μg/mL hyaluronidase (Merck, Darmstadt, Germany) to OD 0.6–0.8 before dilution with TH and 70% glycerol for freezing in 2.5 mL aliquots.

### 2.2. Preparation of Recombinant GAS Antigens

50 mL pellets of *E. coli* containing recombinant the protein plasmids previously described [19] were thawed and suspended in wash buffer (50 mM Tris-HCl pH 7.5, 500 mM NaCl, 20 mM imidazole) containing 1 mg/mL lysozyme and 40 μg/mL DNase I and 1X BugBuster (Merck). Cells were lysed with Lysing Matrix B 0.1 mm silica spheres (MPBioMedicals, Eschwege, Germany) in a FastPrep, incubated with rotation at RT for 30 min, and incubated at 37 °C for 30 min before the soluble fraction was harvested at 17,000× *g* 10 min. The soluble fraction was filtered through a 0.2 μm filter before it was applied to a 1 mL HisTrap-HP column (Cytiva, Marlborough, MA, USA) and purified with an AKTA FPLC system. Protein was eluted with 50 mM Tris-HCl pH 7.5, 500 mM NaCl, 250 mM imidazole, and fractions containing protein were further purified by size-exclusion chromatography on a 16.60 Sephacryl S 300 gel filtration column (Cytiva) in GF buffer (50 mM Tris-HCl pH 7.5, 150 mM NaCl). Fractions containing pure protein were pooled and concentrated using spin columns with a 3 KDa MWCO (Merck). Endotoxin was removed from proteins by Pierce High-Capacity Endotoxin Removal Resin (ThermoFisher Scientific, Waltham, MA, USA), and the proteins were tested by LAL for the presence of endotoxin < 24 IU/mL.

### 2.3. Animals and Immunisations

Six- to eight-week-old BALB/c mice (n = 5) were immunized with 10 μg of each protein antigen (SpyCEP, Cpa, Mac(IdeS) and MalE; 40 μg total dose). For subcutaneous immunization, antigens were either adsorbed individually onto 100 μg/dose (total, 25 μg per antigen) Alhydrogel (Alum) or combined without Alum. For mucosal immunization, antigens were combined with either 10 μg of cholera toxin B subunit (CTB, Merck) or carbonate buffer alone. The sublingual dose was 20 μL; the intranasal dose was 10 μL (5 μL per nostril); and the subcutaneous dose was 200 μL. Animals were anaesthetized in an isoflurane chamber for 15 min for all procedures. The immunogen was deposited slowly and carefully onto the underside of the tongue or in the nostril using a micropipette. Once the entire dose was dispensed, the mouse was returned to the induction chamber and placed in a dorsal uppermost position with its head straightened to ensure that the airways were not obstructed. The mice remained under anesthesia for a further 30 min before they were allowed to recover. In total, 5 doses were given, each a week apart. Sera samples were collected on Day 0, Day 21 (after 3 immunizations) and Day 36 (one week after 5 immunizations). Fecal pellets were collected on Day 21 and Day 36, and washes from intestines were collected at termination. Animal studies were conducted according to the UK Home Office (Scientific Procedures) Act 1986.

### 2.4. Sample Processing

Collected blood was spun at 17,000× *g* for 10 min, and serum was collected for freezing at −20 °C in small single-use aliquots. Fecal samples and intestinal washes were all collected in IgA preservation buffer (2% FCS, 1× protease inhibitor (Thermo)). Feces and intestinal samples were weighed and broken up prior to incubation at 4 °C for 3 h with rotation. Samples were spun at 17,000× *g* for 10 min and stored in small-use aliquots at −20 °C. Fecal and intestinal samples were normalized to 50 mg/mL in IgA preservation buffer for ELISA analysis or in HaCaT buffer on experimental days.

### 2.5. ELISA to Measure IgA and IgG Antibody Titers

MaxiSorp 96-well ELISA plates (Nunc, Roskilde, Denmark) were coated with 50 μL/well of 0.5 μg/mL of each recombinant protein at 4 °C overnight. Plates were blocked with assay diluent (AD: 1% BSA, 0.3% Tween-20, 1× PBS) for 1 h at 37 °C before the addition of 100 μL sample dilutions as indicated, diluted in AD. Plates were incubated for 2 h at 37 °C before the addition of secondary antibodies (goat anti-mouse IgG-HRP 1:20,000 (Sigma, A9044) or goat anti-mouse IgA-HRP (1:2000 Sigma, 1:3000 Abcam where indicated)) for 1 h at 37 °C. Reactions were detected with 100 μL TMB (3,3′,5,5′-Tetramethylbenzidine, Thermo) for 15 min before quenching with 100 μL 1 M sulphuric acid. All steps were separated by 7× washes with PBS-T (1× PBS, 0.05% Tween-20). Optical density was measured at 450 nm on aMolecular Devices V Max kinetic microplate reader and data obtained using SoftMax Pro Version 7.1. The antibody titer for IgG was determined as the serum dilution giving an OD of 0.5, as calculated from interpolation of a sigmoidal titration curve in GraphPad Prism 9. The endpoint titer for IgA was calculated as the sample dilution giving an OD great than 3 standard deviations over the mean of the blank measurements. Data are presented for individual mice and as the GMT of the group. For GMT calculation and statistical analysis, negative samples were given an arbitrary titer of 1.

### 2.6. Immunofluorescence Staining and Flow Cytometry Analysis

Cultures of GAS were grown to log phase (OD 0.4–0.8) and harvested after centrifugation for 5 min at 4000× *g*. Bacteria were suspended in PBS with 10% goat serum and incubated for 20 min at RT. For each test, 1 mL of suspension was harvested before re-suspension in 100 μL staining buffer (0.1% BSA, 10% goat serum, PBS) with and without 1:5 dilution of immune mouse serum. Samples were incubated for 1 h at 4 °C and then washed with 0.1% BSA-PBS. Cells were incubated with 100 μL goat-anti-mouse-IgG-FITC (ThermoFisher Scientific, A16067) diluted 1:3000 in 0.1% BSA-PBS for 45 min at 4 °C. Cells were washed with 0.1% BSA-PBS and suspended in 0.5 mL of fixative (2% formaldehyde, 50% PBS). Sample data was collected on a BD FACS Canto II flow cytometer, with further analysis carried out using FlowJo Software V10.7.1. Mean fluorescence intensity (MFI) was calculated for all groups.

### 2.7. Opsonophagocytosis Assay (OPA)

HL60 cells were routinely grown in HB (RPMI, 10% FCS, 1% l-glutamine) at 37 °C with 5% CO_2_, and 4 × 10^5^ cells/mL were differentiated in HB containing 0.08% DMF (N,N-dimethylformamide) for 5 days. Cells were harvested at 300× *g* for 5 min and washed with 1× HBSS without salts and then 1× HBSS with salts before suspension in OBB (1× HBSS with salts, 5% FCS, 1% gelatin) to 1 × 10^7^ cells/mL. Frozen GAS cultures were thawed quickly, harvested by centrifugation at 12,000× *g* for 2 min, washed in 1 mL 1× HBSS with salts and suspended in 0.5 mL OBB. Suspensions were diluted in OBB to a concentration yielding 60–80 colony-forming units (CFU) when the cells were incubated with complement controls under assay conditions. Sera-free control reactions were set up with active and heat-inactivated complement (Pel Freeze, Rogers, AR, USA) to distinguish non-specific killing from complement, such that only assays where <35% non-specific killing was observed were accepted. 10 μL GAS were incubated with 20 μL buffer or heat-inactivated sera for 30 min at RT at 700 rpm before the addition of 50 μL HL60s with baby rabbit complement (40 μL HL60, 10 μL complement to a final assay dilution of 2%). Reactions were incubated for a further 90 min at 37 °C with 5% CO_2_ at 200 rpm before they were placed on ice for 20 min. Reactions were mixed well, and 5 μL was drip-plated on TH agar. The CFUs were counted the following day. Experiments were set up in triplicate. One-way ANOVA was performed to compare the results to results from a non-immune-sera control using GraphPad Prism.

### 2.8. HaCaT Cell Binding Assay

HaCaT cells were routinely grown in DMEM with high glucose, 10% FCS and 1% L-glutamine at 37 °C with 5% CO_2_. Next, 24-well plates were seeded with 1 × 10^5^ cells/mL and grown at 37 °C with 5% CO_2_ for 3 days to confluence. HaCaT cells were counted in two wells to determine input Multiplicity of Infection (MOIs). Frozen GAS cultures treated with hyaluronidase were thawed, spun at 4,000× *g* for 5 min and washed in assay buffer (1% FCS, 1× HBSS) before dilution to an MOI of ~20. To test the blocking activity of immune sera and mucosal samples, GAS were diluted and pre-incubated for 30 min at RT with rotation with sera (1/100 dilution), mucosal samples (1/2 dilution) or assay buffer alone before they were added to HaCaT cells. Mucosal samples were filter-sterilized, and all immune samples were heat-inactivated at 56 °C for 30 min prior to use. Culture media was removed from HaCaT confluent wells, and 1 mL diluted GAS with or without immune samples was added to assay wells in triplicate. Next, 1 mL GAS was added to growth-control wells in the absence of HaCaT cells. Plates were spun at 800× *g* for 5 min and incubated at 37 °C with 5% CO_2_ for 1 h. Supernatants were then removed from the assay wells, and the monolayers were washed 3× with 1× PBS. Next, 250 μL trypsin-EDTA was added to wells and the wells were incubated for 15 min at 37 °C. Cells were suspended completely with the addition of 750 μL assay buffer and transferred to centrifuge tubes. Samples were harvested by centrifugation at 300× *g* for 5 min, and the supernatant was discarded. Pellets were suspended in 1 mL PBS and serially diluted for a 5 μL drip on TH agar plates. Input CFUs and growth controls were also diluted, and 5 μL were plated on drip plates. Experimental MOI was calculated from input CFUs; growth index was calculated from comparison of input and growth controls; percentage binding was calculated compared to growth controls and percentage blocking was compared to a no-sera control. One-way ANOVA was performed for immune samples compared to the PBS group or between adjuvanted and non-adjuvanted groups using GraphPad Prism.

### 2.9. IL-8 Cleavage Assay

An IL-8 DuoSET ELISA kit (RnD systems Biotechne, Minneapolis, MN, USA, DY208-05) was adapted to measure IL-8 cleavage activity of SpyCEP as follows. GAS cultures were grown to mid-log (OD 0.6–0.8) in THY broth and harvested after centrifugation at 4000× *g*. Supernatants were filter-sterilized, then diluted 1/40 with 125 pg/mL IL-8 standard in a total volume of 10 μL and incubated at 37 °C for 16 h. To evaluate the ability of immune sera to block IL-8 cleavage by GAS supernatants, supernatants were pre-incubated with sera dilutions for 1 h 37 °C prior to addition of IL-8. MaxiSorp 96-well ELISA plates (Nunc) were coated with 100 μL/well of 4 μg/mL capture antibody at RT overnight. Plates were blocked with 300 μL AD at RT for 1 h, and the prepared samples described above were diluted to 100 μL and added to plates for incubation at RT for 2 h. Next, 100 μL detection antibody was added and plates incubated at RT for 2 h before addition of 100 μL streptavidin at RT for 20 min. All steps were separated by 3× washes with PBS-T. Reactions were detected with 100 μL TMB for 15 min at RT and stopped with 100 μL 1 M sulphuric acid. Absorbance was read at 450 nm on a SoftMax Pro plate reader. One-way ANOVA was performed to compare IL-8 cleavage in the presence of immune sera with cleavage in the PBS control group and between adjuvanted and non-adjuvanted groups.

### 2.10. IgG Cleavage Assay

MaxiSorp 96-well ELISA plates (Nunc) were coated with 1:20,000 dilution of goat anti-human Fab (Sigma) overnight at 4 °C. Next, 10 ng of recombinant Mac/IdeS were pre-incubated with sera as indicated at 37 °C for 1 h before the addition of 100 ng human IgG (Sigma) in assay diluent (AD: 1% BSA PBS-T 0.3% Tween-20), then further incubated at 37 °C for 3 h. ELISA plates were blocked for 1 h in AD before the addition of assay samples for 1 h and incubation with 1:20,000 goat anti-human Fc-HRP (Sigma) for 1 h. All steps were conducted at 37 °C and separated by three washes with PBS-T (0.05% Tween-20). Reactions were detected with 100 μL TMB incubated at RT for 15 min before stopping with 100 μL 1 M sulphuric acid. Plates were read at 450 nm on a SoftMax Pro plate reader. Signal was compared with the signal from no-sera assay conditions. One-way ANOVA was performed to compare IgG cleavage in the presence of immune sera with the PBS control group and between adjuvanted and non-adjuvanted groups.

## 3. Results

Four recombinant proteins (SpyCEP, Cpa, Mac(IdeS) and MalE) were used to immunize mice through sublingual (SL), intranasal (IN) or subcutaneous (SC) routes with and without adjuvant (+/−). Serum IgG and mucosal IgA levels were determined by ELISA, and the functional activities of the induced response were evaluated using antigen-specific in vitro assays.

### 3.1. Subcutaneous and Mucosal Immunization Generate Robust Systemic Immune Responses

A substantial serum IgG response was induced against all four recombinant antigens after 5 doses by both mucosal and subcutaneous immunization (Figure 1), except for MalE, for which a poor IgG response was induced following SL immunization That response was not statistically significantly higher than the response of the PBS negative-control group. For SpyCEP and Cpa, the response was comparable for the IN and SC routes of administration (Figure 1A,B). The response for SL immunization was slightly lower, significantly above the response of the PBS group only for Cpa (SpyCEP GMT 4231, *p* = 0.051; Cpa GMT 8143, *p* = 0.0305). For Mac and MalE, mucosal immunization elicited slightly lower IgG responses than did SC immunization (Figure 1C,D). IN immunization generated a significant increase in IgG above the PBS group for MalE, but the difference was not significant for Mac (Mac, *p* = 0.0577; MalE, *p* = 0.0492). SL generated a small IgG response for both Mac and MalE, but it was not significantly greater than the response in the PBS control group (*p* > 0.05).

Induction of an IgG response following 5 mucosal immunizations required the presence of CTB adjuvant, although a small IgG response could be observed to SpyCEP and Cpa administered by the intranasal route without adjuvant in 4/5 mice (GMT 50 and 170, respectively, for responding mice; Figure 1A,B). The IgG response that followed subcutaneous immunization was not dependent on the presence of adjuvant after 5 immunizations with any antigen except MalE, for which the presence of adjuvant was necessary to induce an IgG response that was statistically significantly greater than that in the PBS control group (SC+ GMT 2712, *p* = 0.0005; SC− GMT 42, *p* = 0.0725; Figure 1D).

When the responses following 3 and 5 doses of immunization were assessed for SC immunization, it was found that for SpyCEP, Cpa and Mac, alum adjuvant was necessary to induce a higher initial immune response after 3 doses, but not after 5 doses. After 5 doses, by D36, the response in groups with and without adjuvant were comparable. Meanwhile, the MalE IgG response was dependent on all 5 doses even in the presence of alum. For mucosal immunizations, 5 doses were necessary to induce a substantial IgG response, with the exception of intranasal immunization for SpyCEP and Cpa, which showed a significant IgG response over PBS after 3 immunizations (*p* = 0.0044 and *p* = 0.0076 respectively). There was still a significant increase in IgG titer between 3 and 5 doses despite this initial strong response (SpyCEP *p* = 0.0189; Cpa *p* = 0.0035). Cpa also induced a significant level of IgG after only 3 doses by the SL route compared to the PBS control group (GMT 482, *p* = 0.0125), which then further increased significantly after two more doses (*p* = 0.0127). The final IgG titer was greatest for SpyCEP and Cpa, with lower titers observed for Mac and MalE.

### 3.2. Mucosal Immunization Can Generate a Secreted IgA Response after 5 Doses

#### 3.2.1. IgA Response in Intestinal Samples

Due to their large volume size, intestinal washes were collected and used to measure distal IgA production and secretion against all four antigens following mucosal and SC immunizations. Data presented in Figure 2 shows that intranasal immunization with adjuvant generated the most significant IgA response compared with the PBS control group for all antigens (SpyCEP: GMT 199, *p* = 0.0001; Cpa: GMT 692, *p* = 0.0003; Mac: GMT 19, *p* = 0.0004; MalE: GMT 4, *p* = 0.047). Meanwhile, sublingual immunization with adjuvant produced a significant IgA response for Cpa and a small but significant IgA response for Mac (GMT 256, *p* = 0.0013; GMT 2, *p* = 0.038, respectively; Figure 2B,C) but did not produce a significant IgA response for SpyCEP or MalE (Figure 2A,D). Subcutaneous immunization was only able to generate a small but statistically significant IgA response for Mac (GMT 4.7, *p* = 0.003, Figure 2C). The IgA response was completely dependent on the presence of adjuvant and while not all animals responded to sublingual and subcutaneous immunization, intranasal immunization demonstrated a much more consistent response between animals, particularly for SpyCEP and Cpa.

#### 3.2.2. IgA Response at D21 and D36

Fecal samples taken at D21 and D36 after three and five immunizations respectively were processed to assess the production of mucosal IgA for SpyCEP and Cpa to determine whether all five doses are required for the IgA response following mucosal immunization (Appendix A). Only five but not three immunizations by the intranasal route with adjuvant induced a significant fecal IgA response for both SpyCEP and Cpa (SpyCEP, *p* = 0.002; Cpa, *p* = 0.042, compared to PBS group). Sublingual immunization did not induce any fecal IgA response significantly above the PBS group after three or five immunizations. This demonstrates that at least for IN immunization, five doses are needed to induce a mucosal IgA response at distal mucosal sites.

### 3.3. Antibodies from Immunisations Are Able to Bind to and Kill GAS Cells

#### 3.3.1. Surface Binding of IgG Antibodies to GAS Cells

GAS cells were incubated with sera samples from mice immunized SL, IN and SC in the presence or absence of adjuvants and assessed for surface binding of IgG by flow cytometry. As Figure 3A shows, there was a clear shift in FITC signal using immune sera from adjuvanted groups compared to the PBS control group. This is further verified with mean fluorescence intensities (MFI) values that are all 200 units above the PBS group (Figure 3A). Sera from non-adjuvanted IN and SC groups still showed a shift in MFI compared with the PBS group but to a lesser extent than their adjuvanted counterparts. This correlates with the ELISA data from Figure 1.

#### 3.3.2. Killing of GAS Cells by Opsonophagocytosis

An opsonophagocytosis assay was conducted with heat inactivated sera, baby rabbit complement and differentiated HL60 cells and assessed for killing activity. Immune sera from intranasal and subcutaneous immunizations with adjuvant and subcutaneous immunization without adjuvant showed high killing activity similar to a positive control serum sample from our previous study (27–30% killing, Figure 3B). However, this was only statistically higher than the normal mouse serum control for sera from the intranasal group with adjuvant (*p* = 0.0496) but not for sera from the subcutaneous groups (*p* > 0.05). Very low, variable and non-significant results were obtained with immune sera from sublingual immunization.

### 3.4. Mucosal and Subcutaneous Immunisation Can Block Functional Activity of Proteins

#### 3.4.1. Blocking Binding of GAS Cells to HaCaT Monolayers

A GAS-keratinocyte binding assay using HaCaT cells has been developed and used to investigate the ability of the immune sera and mucosal secretions generated following immunization to interfere with GAS binding. GAS-HaCaT cells binding was first tested in the presence of immune sera samples, with results demonstrating the ability of immune serum antibodies from SL or IN immunized mice in the presence of adjuvant to almost entirely block GAS binding to HaCaT cells (Figure 4A; SL, 68%; IN, 87%). Sera from mice immunized SC in the presence or absence of adjuvant were also able to block GAS binding (Figure 4A; SC+, 89%; SC−, 83%). The percentage blocking was significantly greater than blocking by serum from the PBS negative control group (28%) for all three immunization routes (SL+, *p* = 0.0139; IN+, *p* = 0.0003; SC+, *p* = 0.0002; SC−, *p* = 0.0006). For mucosal immunization this blocking was dependent on the use of adjuvant as immune sera from non-adjuvanted groups were not able to block any binding of GAS to HaCaT cells above the negative control, whereas for subcutaneous immunization there was no significant difference between the adjuvanted and non-adjuvanted groups. Blocking of GAS binding was also tested with mucosal (intestinal) secretion samples from mice immunized in the presence of adjuvant that had demonstrated an IgA response by ELISA (Figure 4B). Results showed that intestinal washes from both SL and IN active immunization groups had greater blocking activity than secretions from the PBS group (SL+, 68%; IN+, 77%, PBS, 50%), however this was only statistically significant for secretions from intranasal immunization (*p* = 0.0293).

#### 3.4.2. SpyCEP Activity Can Be Neutralized with Immune Sera

SpyCEP is an IL-8 degrading protease and its activity can be monitored in late-log supernatants from GAS cultures [20]. Neutralization of GAS supernatant-mediated IL-8 cleavage by immune sera generated in our study was evaluated by ELISA. Positive control sera from our previous study shows effective neutralization of SpyCEP activity through a reduction in IL-8 cleavage [19]. Subcutaneous immunization induced the most effective serum-mediated neutralization of SpyCEP activity in GAS supernatants, with only ~30% cleavage of IL-8 (66% neutralization, Figure 5A) when immune sera were mixed with GAS supernatant and IL-8. Immune sera from intranasal immunization with adjuvant and subcutaneous immunization without adjuvant mediated a 30 and 29% reduction in IL-8 cleavage compared to GAS supernatants alone, respectively. This level of reduction was significant (*p* < 0.0001) for all samples compared to the activity of the serum from the PBS control group. Meanwhile, sera from sublingual immunization and low-immune response samples (SL and IN without adjuvant and PBS control) showed less than 5% reduction in cleavage of IL-8, with a significant difference in activity between adjuvanted and non-adjuvanted groups (SL, *p* = 0.015; IN, *p* < 0.0001; SC, *p* < 0.0001). A significant correlation was observed between the level of neutralizing activity and the anti-SpyCEP IgG titer (R^2^ = 0.9762), as shown in Appendix A.

#### 3.4.3. Mac(IdeS) IgG Cleavage Activity Can Be Neutralized in a Titer Dependent Manner

Mac is an enzyme produced by GAS that can cleave human IgG (huIgG) at the hinge region. We have previously shown by western blot that Mac anti-sera can neutralize this cleavage activity [19]. An ELISA was developed to distinguish between intact and cleaved huIgG such that neutralization of Mac activity with immune sera could be measured for improved comparison. The lower the percentage cleavage of IgG the higher the neutralizing effect of immune sera against Mac. Recombinant Mac was incubated with huIgG in the presence or absence of sera, before performing a capture ELISA to detect full length huIgG. Mac alone with no sera shows ~65% cleavage of IgG, and as Figure 5B shows, samples containing no anti-Mac antibody (mucosal immunizations without adjuvant) demonstrate similar levels of IgG cleavage, with no neutralization of activity. This was markedly reduced with samples containing anti-Mac antibodies. Sera from subcutaneous immunization mediated the highest neutralization of Mac activity (84%), however the neutralization activity of immune sera from sublingual and intranasal immunization groups with adjuvant was also significantly greater than the serum from PBS control (60 and 78% neutralization respectively, *p* < 0.0001 for all). This activity could be maintained when diluted samples were used until a 1/50 dilution before a loss of neutralization activity, and when R^2^ analysis was performed to compare IgG cleavage to the anti-Mac IgG GMT for each group, a strong correlation was observed (R^2^ = 0.8355, Appendix A).

## 4. Discussion

GAS infections and resultant autoimmune diseases are a huge global burden with no current prevention beyond antibiotic therapies. Economically, milder infections such as pharyngitis and impetigo can cost up to $2900 per episode, while cases of severe RHD can cost up to nearly $40,000 depending on the setting, a value similar to the economic burden of certain cancers [21]. There is therefore a great need for vaccines to prevent infections and reduce the overall impact of these GAS infections to improve DALYs and economic burden, particularly in LMICs. To combat this, the WHO have produced a roadmap to GAS vaccines, highlighting the need to prevent the mucosal infections pharyngitis and impetigo as a precursor for prevention of RHD [6]. We have previously shown that a formulation of protein antigens that represent stages of GAS infection are as effective as M-protein immunizations at induction of functional immunity against GAS [19]. These proteins represent colonization (Cpa), saliva survival (MalE), invasion and suppression of the immune system (SpyCEP, Mac/IdeS), antigens that are key elements of GAS infection. All but Cpa are conserved proteins and were used as an indication of stages of GAS infection that could be targeted by effective vaccines. In this current study, we took the same formulation and investigated the effects of different routes of immunization, particularly to stimulate mucosal immunity. It has previously been shown that human IgG is more effective at opsonizing and killing GAS than human IgA, but despite this, only IgA was able to block intranasal infection and survival of mice following passive immunization [13]. It was therefore hypothesized that secretory IgA alone protects against the initiation of infection, however a serum response must be necessary to protect against other aspects of infection, as there was not complete survival of animals. We were therefore interested in stimulating both secretory IgA and systemic IgG through a single route of immunization and comparing functionality of the induced immune response.

The entry point for GAS is generally through the mucosa, with the oropharyngeal site as key for mucosal immunity. In this study, both sublingual and intranasal routes of immunization were examined to compare with classic subcutaneous immunization as a means to stimulate mucosal immunity whilst also generating a systemic immune response. The sublingual mucosa is devoid of M cells and does not have an organized lymphoid structure, with dendritic cells playing an important role in antigen uptake before migration to the draining lymph nodes for initiation of an immune response [22]. Meanwhile, there is a richness of lymphatic tissue in the nasal cavity with the nasal associated lymphatic tissues (NALT), but this area has a smaller capacity for vaccine uptake [12]. Further, care needs to be taken with intranasal immunization to avoid irritation and neurotoxicity due to the risk of “nose-to-brain” delivery through the intranasal route [12]. In our study, both routes were superior to subcutaneous immunization for the ability to stimulate both serum IgG and mucosal secretory IgA. However, intranasal immunization induced generally higher titers and the most consistent response between animals within a group compared with sublingual immunization, which showed a large spread of IgA titers between animals, with some showing a low or no response at all. This is likely due to the method of immunization where in the sublingual immunization there is a risk of antigen wash out due to saliva flow and a lack of control of the dose adsorbed [22]. Advances in mucoadhesive formulations could increase the antigen contact time with the sublingual mucosa and improve immune responses through the sublingual route [22]. Meanwhile, subcutaneous immunization was mostly ineffective at mounting an IgA response, which is consistent with the literature. Oral and nasal IgA would give a clearer indication of the local immune response in the mouth and nose but unfortunately sampling for this study from the mouth and nose was very limited and showed high background and no consistent measurement of IgA. Therefore, intestinal washes were used as an indicator of mucosal immunity.

While it is difficult to compare antibody titers between protein antigens due to the potential for inconsistencies in coating of plates, generally SpyCEP and Cpa generated the highest IgG titers for all routes, with Mac and MalE showing a lower IgG response. This was further observed with the IgA response, where intranasal immunization generated an immune response at least one log lower for Mac and MalE than for SpyCEP and Cpa. This was surprising for MalE, as though it showed a lower IgG response to immunization when we previously measured the response in IVIg [19], we had hypothesized that this low IgG response was potentially due to its role in survival in saliva, and that an IgA response may be more representative of the immune response to target this antigen. This was not the case however and demonstrates that though a protein may be important in a particular stage of infection it may not be an effective vaccine target or may require a higher dose to stimulate a significant immune response. In the context of a recombinant protein formulation the charge of the molecules may affect their ability to induce a mucosal immune response, with proteins of a lower pI having a more favorable positive charge in physiologically neutral pH conditions to bind to negatively charged mucus more effectively. Mac and MalE both have higher predicted pI values (6.1 and 5.8 respectively), whereas SpyCEP and Cpa had lower predicted pI values (5 and 5.3 respectively) and induced a higher mucosal immune response.

In addition to the mucosal routes inducing superior secreted IgA responses than subcutaneous immunization, it should particularly be noted that these routes also induced excellent serum IgG responses. Immune sera from all three routes of immunization with adjuvant showed similar IgG binding to GAS cells by flow cytometry. Due to the surface localization of Cpa and our previous study comparing combination and individual immunizations this was likely mostly driven by the anti-Cpa response, which showed the most consistent serum IgG response between immunization routes [19]. Similarly, immune sera from all three routes mediated effective blocking of GAS binding to HaCaT cell monolayers, with immune sera from the intranasal group being the most superior across sera and mucosal samples for significant blocking, though all three routes where an effective IgG and IgA response were generated showed excellent blocking ability compared with low antibody response groups. Differences became evident in the OPA where both intranasal and subcutaneous groups were able to induce antibodies that effectively mediate the killing of GAS cells. Meanwhile, lower and varied activity was measured using immune sera from the sublingual route with adjuvant, likely due to the lower and more varied response between mice against each antigen. Of particular importance was the evidence of the functionality of the IgG antibody responses for SpyCEP and Mac, two virulence factors that are of greater importance in systemic infection and would require a robust IgG response. There was excellent functional neutralization of SpyCEP and Mac cleavage activities by immune serum antibodies, with lower neutralization with sera generated from mucosal routes compared with the subcutaneous route attributed to lower titers of IgG. This trend was linear and demonstrated that there was no loss of relative functionality, but the reduction was purely from lower antibody titers. These IgG titers were consistently lower than those from subcutaneous immunization, particularly for SpyCEP and Mac where sublingual immunization was shown to induce one log lower IgG titer than subcutaneous immunization for both antigens. It would therefore be of interest to investigate dosing and adjuvant conditions to obtain comparable sera IgG titers to subcutaneous immunization. There should also be a focus on investigating the type of T cell response induced with the used of different routes of immunization and adjuvants and the impact on antibody functionality and protection.

In this study we used five weekly doses of vaccine as this had been shown in preliminary work and in a study of GBS vaccine mucosal delivery to mount good immune responses [11]. To assess the benefit of five doses for our combination vaccine, we tested sera and fecal samples for IgG and secreted IgA responses respectively after three and five immunizations. Mucosal immunization particularly benefitted over subcutaneous immunization with five doses generally showing a more significant increase in IgG titer compared to three doses for most antigens. This was less evident with the IgA response, possibly due to the lower titers and spread of response between animals, however it could be observed that only the five doses produced a significantly high titers of IgA, particularly for Cpa. Fecal samples showed less consistent IgA measurements than intestinal washes and had higher background, which limited the analysis. This is already a reduction on dosing compared with a study where lipoteichoic acid was administered with CTB by the intranasal route through eight immunizations over two weeks [14]. Reductions in boosting improves uptake of vaccines by the population so ensuring the highest response balanced with a low dose requirement is important for a cheap, sought after and effective vaccine.

Our study and others have demonstrated that unlike subcutaneous routes, adjuvant is essential to elicit an immune response following mucosal delivery of a subunit-based vaccine [14]. In the current study we used the well-established CTB as the mucosal adjuvant [14,16]. This has been proven safe in humans and is a component of and adjuvant in the Dukoral oral vaccine [23]. However, there are several novel adjuvants that could be utilized to improve the immune response, such as those that target pattern recognition receptors such as Toll-like receptors including monophosphoryl lipid A (MPL) that is formulated with Alum to produce the AS04 adjuvant used in Cervarix against human papilloma virus [23] or MPL formulated with *Quillaja Saponaria* Molina to produce AS01, which can increase antigen specific CD4+ T cell response the liposomal structures of which may be useful for mucosal delivery [24]. Whole cell vectors such as *Lactococcus lactis*, or cell-like structures could potentially bypass the need for adjuvant systems. A study with *L. lactis* expressing M-protein units was able to produce serum IgG and pharyngeal IgA without the need for adjuvant through the intranasal route [25]. Oral gavage of *L. lactis* expressing pilin proteins was also able to stimulate a bronchial IgA response as well as a serum IgG response, demonstrating the benefit of cellular structures in mucosal immunity [15]. Similarly, some vectors resembling cell-like structures such as generalized modules for membrane antigens (GMMA), liposomes, microparticles and virus-like particles (VLP) show potential benefits to delivery. A study with VLPs expressing J8 were shown to elicit an effective serum IgG and salivary IgA response that was superior to the response from subcutaneous immunization, however they also demonstrated that this salivary IgA response was further improved by cholera toxin adjuvant [17]. Similarly, a liposomal formulation with cationic adjuvant formulation 01 (CAF01) with C5a peptidase saw a significant increase in IgA production in the lungs compared with alum or CpG adjuvants, particularly with delivery via the subcutaneous route followed by intranasal delivery [26]. Therefore, a dependence on the use of an adjuvant may be necessary for optimal mucosal protection from infection even with cell-like structures. A recent study demonstrated that priming by intramuscular immunization with CAF01 followed by intranasal immunization without adjuvant induced effective mucosal protection [8]. Formulation will certainly be a key consideration in designing mucosal vaccines in the coming years to ensure efficient trafficking of antigens and targeting of both the cellular and humoral branches of the immune system. More focus on the effect of stimulating the different types of T cell response in GAS vaccine design is needed, and an assessment of memory responses generated from different methods of immunization. Further, selecting the appropriate adjuvant for the required immune response will be important, especially should any correlates of protection be uncovered in the coming years. Studies have shown that the sublingual route with CTB or heat-labile toxin can generate IgA at mucosal sites and serum IgG to protective levels against other species of Streptococci [10,11]. Several studies have demonstrated intranasal immunization can generate a mucosal immune response against GAS [14,16,25], but ours is the first study to show that a response can also be generated through immunization with a subunit vaccine by the sublingual route, which has potential safety benefits over intranasal immunization.

Finally, an indication of cost must be considered. Recombinant protein vaccines are relatively cheap to produce, but other options such as GMMA could be considered as a low-cost and potentially adjuvant free system for vaccine delivery. Low dose and low boost factors need to be considered so an effective route to producing a strong mucosal IgA and systemic IgG needs to be further optimized, either through alternative platforms such as GMMA or cell-like structures, or improved adjuvant systems.

## Figures and Tables

**Figure 1 vaccines-11-01724-f001:**
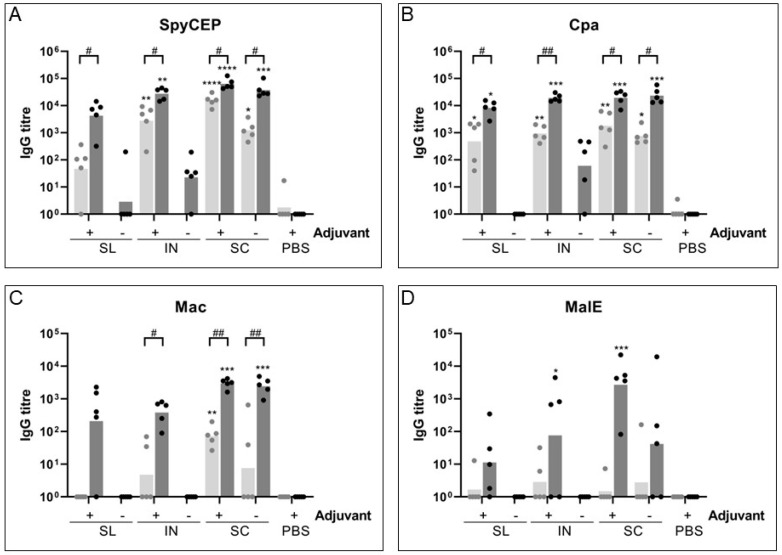
**Sera IgG response to vaccine antigens.** Test bleeds from Day 21 (light grey) and terminal bleeds from Day 36 (dark grey) were collected and tested by ELISA for an IgG response to each vaccine antigen in individual mice for sublingual (SL), intranasal (IN) or subcutaneous (SC) administration or for the PBS control with (+) or without (-) adjuvant. Kruskal-Wallis ANOVA was performed to compare groups with the PBS control. * *p* < 0.05; ** *p* < 0.01; *** *p* < 0.001; **** *p* < 0.0001. One-way ANOVA was performed to test for differences between the D21 and D36 response. #, *p* < 0.05; ##, *p* < 0.01.

**Figure 2 vaccines-11-01724-f002:**
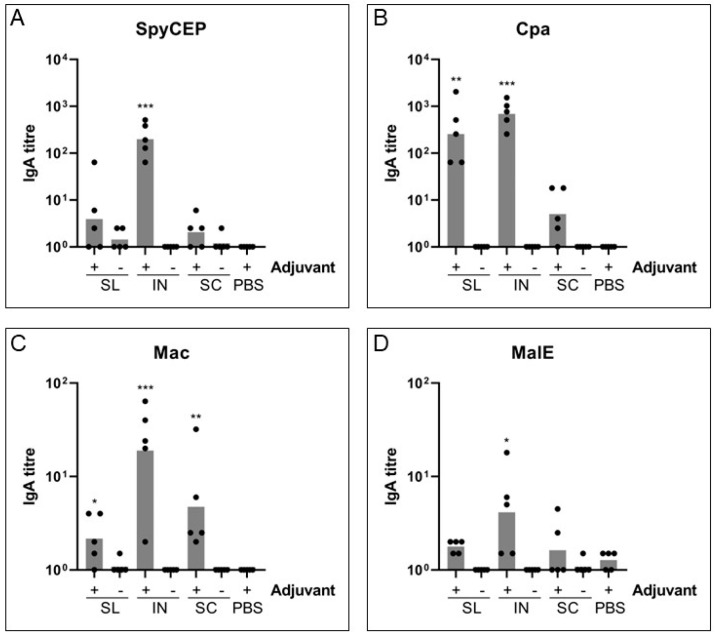
**Mucosal secreted IgA response to vaccine antigens.** Distal secreted IgA was detected in terminal intestinal washes from individual mice for sublingual (SL), intranasal (IN), subcutaneous (SC) and PBS control with (+) or without (-) adjuvant. Kruskal-Wallis ANOVA was performed to compare groups with the PBS control. * *p* < 0.05; ** *p* < 0.01, *** *p* < 0.001.

**Figure 3 vaccines-11-01724-f003:**
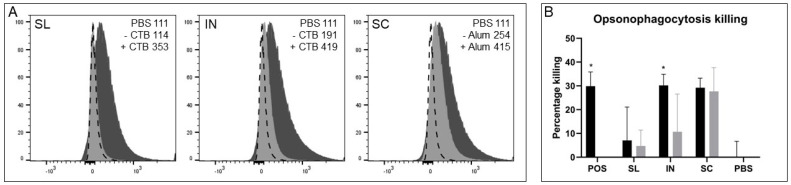
**Antibodies can bind to GAS cells and promote killing by OPA.** GAS cultures were incubated with sera from sample groups and investigated for IgG binding by flow cytometry (**A**) and killing activity (**B**). POS, positive control; SL, sublingual; IN, intranasal; SC, subcutaneous; PBS, negative control group. (**A**) Immunostaining was conducted with sera samples and analysed by flow cytometry. Specific binding was demonstrated with a shift in FITC fluorescence. Mean fluorescent intensity (MFI) is shown numerically for each panel with (+) and without (-) adjuvant indicated. Dashed line, PBS control; light grey, without adjuvant; dark grey, with adjuvant. (**B**) Opsonophagocytosis assays (OPA) were conducted to demonstrate killing activity of sera in the presence of DMF-differentiated HL60 cells and baby rabbit complement. Percentage killing was calculated by CFU relative to a non-immune standard normal mouse sera control. Adjuvanted groups are indicated with black bars and non-adjuvanted groups are indicated with grey bars. Data are presented as the mean % killing +/- standard deviation. One-way ANOVA comparison with non-immune sera is indicated. *, *p* < 0.05.

**Figure 4 vaccines-11-01724-f004:**
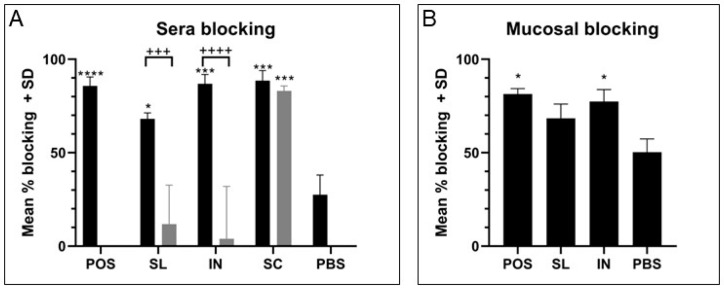
**Antibodies can bind to GAS cells with functional activity.** GAS cultures were incubated with immune sera samples diluted 1/100 (**A**) or mucosal secretions (intestinal) normalised to 50 mg/mL (**B**) and investigated for blocking of binding to HaCaT monolayers. POS, positive control; SL, sublingual; IN, intranasal; SC, subcutaneous; PBS, negative control group. Adjuvanted groups are indicated with black bars and non-adjuvanted groups are indicated with grey bars. One-way ANOVA comparison with PBS control is indicated: *, *p* < 0.05; ***, *p* < 0.001; ****, *p* < 0.0001. One-way ANOVA comparison between adjuvanted and non-adjuvanted controls is indicated: +++, *p* < 0.001; ++++, *p* < 0.0001.

**Figure 5 vaccines-11-01724-f005:**
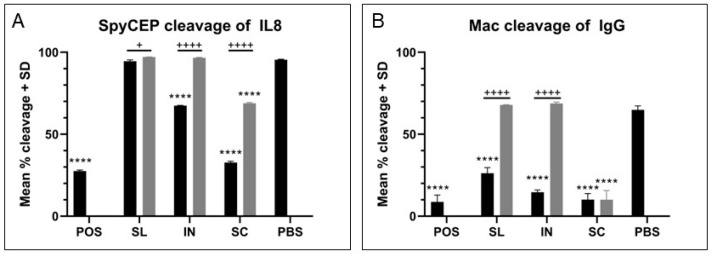
**Antibodies have functional activity against specific vaccine antigens.** Functional assays were conducted for SpyCEP (**A**) and Mac/IdeS (**B**) to test activity of immune sera. POS = positive control. Black bars, adjuvanted groups; grey bars, non-adjuvanted groups. One-way ANOVA significance is indicated above individual bars for comparison with the PBS control (****, *p* < 0.0001), and significance between groups is indicated with bars (+, *p* < 0.05; ++++, *p* < 0.0001). (**A**) An IL-8 cleavage assay was conducted with SpyCEP positive culture supernatants in the presence of sera samples with IL-8 cleavage compared to a no-sera control. Low percentage cleavage indicates high neutralisation of SpyCEP activity by the sample. (**B**) An IgG cleavage assay was conducted with recombinant Mac in the presence of sera samples. Cleavage was compared to a no-sera control as indicated by ELISA. Low percentage cleavage indicates high neutralisation of Mac activity by the sample.

## Data Availability

Data are contained within the article and Appendix A.

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
