# Peer review of "Mucosal Immunization Has Benefits over Traditional Subcutaneous Immunization with Group A Streptococcus Antigens in a Pilot Study in a Mouse Model"

_vaccines, 2023, doi:10.3390/vaccines11111724_

Round 1
Reviewer 1 Report
Comments and Suggestions for Authors
This article makes important contributions to one of the main human bacterial pathogens. The results of immunization in animals arouse interest in clinical research, already recommended by the World Health Organization. All topics in the article are well written, referenced, statistically correct and with very attractive figures. For this reason, I recommend the article for publication.
Author Response
Thank you for taking the time to review our manuscript, and we appreciate your support for our publication.
Reviewer 2 Report
Comments and Suggestions for Authors
This manuscript addresses the urgent need for a vaccine against Group A Streptococcus (GAS), a significant human pathogen without an approved vaccine. The study emphasizes the requirement for a robust immune response, encompassing both systemic and mucosal components, to hinder initial colonization and potential progression to invasive disease. To achieve this, a diverse array of recombinant proteins representing distinct stages of GAS infection was designed. Investigations involving immunization of mice via sublingual, intranasal, and subcutaneous routes, with subsequent measurement of IgG and IgA antibody levels and their functional efficacy through in vitro assays was conducted.
The results demonstrated that sublingual and intranasal immunization, in conjunction with an adjuvant, successfully elicited both systemic IgG and mucosal IgA responses. Conversely, subcutaneous immunization primarily triggered a serum IgG response. Furthermore, the antibodies exhibited substantial functionality by binding to and eliminating GAS cells. They also hindered GAS binding to HaCaT cells, particularly following intranasal and subcutaneous immunizations. Notably, immune sera effectively hindered the cleavage of IL-8 by SpyCEP and IgG by Mac/IdeS.
This study underscores the potential of mucosal immunization in generating an efficacious systemic and mucosal antibody response. It advocates for further exploration and refinement of this approach as a promising alternative to subcutaneous immunization, offering substantial advantages in the prevention of mucosal infections.
Comments
Lines 31-33: Please include reference.
Lines 45 and 46: Mention pharyngitis AND impetigo; toxic shock syndrome AND necrotizing fasciitis.
Line 52: Please elaborate on the statement “many lethal systemic infections do not follow mucosal colonization”.
Line 55: The J8 and p*17 vaccine candidates have also been developed as combination vaccines with a minimal peptide from SpyCEP. These candidates are currently undergoing Phase I clinical trials. Please refer to publication: DOI: 10.1038/s41598-020-80508-6. and DOI: https://doi.org/10.1128/mbio.03537-20
Line 61: GAS vaccine IN delivery has also shown to protect against URT infection. Please refer to: DOI: 10.1128/mBio.03537-20, DOI: 10.1038/srep39274 and https://doi.org/10.1086/505146.
Line 67: Statement on how “none have investigated combination vaccines and most other vaccines focus on serum IgG” is incorrect. Please refer to the above publications.
Line 110: Can the authors confirm that SC immunisation was conducted with/without Alum and IN/SL immunisation was conducted with/without CTB. Please make this clear and reiterate this in the results/figure legends.
Lines 110-130: Sera and faecal pellets were collected during the study. It would have been interesting to collect saliva via 0.1% pilocarpine injection and/ bronchoalveolar lavage (BAL) fluid to assess secretory/mucosal Abs. This would have been a better read out for mucosal immune responses.
Line 155: Some critical, methods related information is missing. For example- What was the dilution of the goat anti-mouse IgG FITC? How the non-specific binding of Abs to GAS (as GAS has Fc receptors) was addressed. Did the authors block the Fc receptors?
Line 170: Start sentence with ‘Ten’ instead of ‘10’.
Line 175: Did authors compare HI complement with A complement to report non-specific killing control for the assay (based on the widely accepted protocols, this should be below 35% NSK). Please also report the source of complement for this assay (i.e. Sigma/Pel Freeze)?
Figure 1: It is hard to distinguish between light grey and dark grey. Please make the colours more distinguishable.
Line 286: Intestinal wash protocol is not listed in the methods.
Figure 2: It would be interesting to see the IgA response in the intestinal washes after concentrating the washes. I assume the IgA is diluted because large volumes of the washes are collected. As mentioned previously the IgA response in saliva and BAL samples would have been more informative. What was the starting dilution of the mucosal washes. Starting at 1:2 dilution and continuing with a 1:2 dilution down the plate will sive more informative data/also detect IgA at low concentrations. Also consider changing the scale in the graph to Log2. As per Figure 1, the colour combination is confusing. b
Figure 3A and B: What GAS strain was used for this study (emm type, animal passaged/not passaged, origin of the strain)? Figure 3A did the authors assess surface binding with mucosal samples? Have they assessed mucosal IgG responses? In Figure 3A, why didn’t they assess GAS binding with Alum adjuvanted/non adjuvanted samples? For Figure 3B, did they dilute the sera to determine non-specific killing, OPK titre and opsonic index? Please refer to manuscript: DOI: 10.1007/978-1-0716-0467-0_26.
Lines 551-555: There is mention of AS04 adjuvant without any mention of the role of T-cells/Th1 in GAS vaccine design. Also, worth mentioning AS01 adjuvant as this is a liposomal formulation and has the potential to be administered IN.
Line 555: Very risky to mention potential whole-cell vaccine design for GAS. Please consider removing or elaborating with a rationale on this.
Line 556: Recent studies with CAF01 show that priming with Ag+CAF01 (SC/IM) followed by pulling with Ag alone (no adjuvant) also induces protective Ab responses in the mucosa and protection. Please refer to publication: DOI: 10.1128/mBio.03537-20.
Comments on the Quality of English LanguageQuality of English language is acceptable.
Author Response
Thank you for taking the time to review our manuscript and provide support to improve the content. We hope we have responded to your comments satisfactorily and made the necessary improvements to the manuscript text. Please see specific responses to comments below in italics.
Lines 31-33: Please include reference.
References have been added.
Lines 45 and 46: Mention pharyngitis AND impetigo; toxic shock syndrome AND necrotizing fasciitis.
This has been corrected.
Line 52: Please elaborate on the statement “many lethal systemic infections do not follow mucosal colonization”.
We have altered this sentence to be more precise.
Line 55: The J8 and p*17 vaccine candidates have also been developed as combination vaccines with a minimal peptide from SpyCEP. These candidates are currently undergoing Phase I clinical trials. Please refer to publication: DOI: 10.1038/s41598-020-80508-6. and DOI: https://doi.org/10.1128/mbio.03537-20
This has been included, though there are several studies that have not been specifically mentioned due to their number as this was not in the scope of the introduction.
Line 61: GAS vaccine IN delivery has also shown to protect against URT infection. Please refer to: DOI: 10.1128/mBio.03537-20, DOI: 10.1038/srep39274 and https://doi.org/10.1086/505146.
This has been added into the manuscript.
Line 67: Statement on how “none have investigated combination vaccines and most other vaccines focus on serum IgG” is incorrect. Please refer to the above publications.
The text has been updated and clarified now for this whole section.
Line 110: Can the authors confirm that SC immunisation was conducted with/without Alum and IN/SL immunisation was conducted with/without CTB. Please make this clear and reiterate this in the results/figure legends.
This has been clarified in the text.
Lines 110-130: Sera and faecal pellets were collected during the study. It would have been interesting to collect saliva via 0.1% pilocarpine injection and/ bronchoalveolar lavage (BAL) fluid to assess secretory/mucosal Abs. This would have been a better read out for mucosal immune responses.
This was our first experiment into mucosal immunity and as such we used intestinal washes as an indicator of mucosal immunity as large volumes were needed for analysis. We are investigating alternative sampling methods for our further studies.
Line 155: Some critical, methods related information is missing. For example- What was the dilution of the goat anti-mouse IgG FITC? How the non-specific binding of Abs to GAS (as GAS has Fc receptors) was addressed. Did the authors block the Fc receptors?
This information is present “diluted 1:3,000”. We have highlighted it in the text for reference. We had used goat serum in the blocking buffer to reduce the non-specific binding of the goat raised secondary antibodies and only compared the shifts to non-immune sera rather than unstained controls to allow for the non-specific binding of antibodies to GAS, as even non-immune sera show a shift in signal. As can be seen from the data this was a suitable analysis as the low antibody responses from active immunisations were the same as the shift from the PBS control group.
Line 170: Start sentence with ‘Ten’ instead of ‘10’.
As all volumes in the manuscript have been referred to numerically we have opted to leave this as a number rather than altering to text, especially to be consistent with the language within the sentence. We hope the reviewer is content with this editorial decision.
Line 175: Did authors compare HI complement with A complement to report non-specific killing control for the assay (based on the widely accepted protocols, this should be below 35% NSK). Please also report the source of complement for this assay (i.e. Sigma/Pel Freeze)?
These were compared and assays only accepted if this was less than 35% NSK. We have included this information in the text along with the complement source.
Figure 1: It is hard to distinguish between light grey and dark grey. Please make the colours more distinguishable.
We have attempted to amend this in the graphs – we wished to be able to distinguish between both bars with/without adjuvant, and the GMT bars with individual mouse points. We hope the contrast is now sufficient.
Line 286: Intestinal wash protocol is not listed in the methods.
This information is already present in the sample processing section of the methods: “Faeces and intestinal samples were weighed and broken up…” We have highlighted this for clarification.
Figure 2: It would be interesting to see the IgA response in the intestinal washes after concentrating the washes. I assume the IgA is diluted because large volumes of the washes are collected. As mentioned previously the IgA response in saliva and BAL samples would have been more informative. What was the starting dilution of the mucosal washes. Starting at 1:2 dilution and continuing with a 1:2 dilution down the plate will sive more informative data/also detect IgA at low concentrations. Also consider changing the scale in the graph to Log2. As per Figure 1, the colour combination is confusing.
Samples were not concentrated further in case there was any loss of material, instead they were made to as high a concentration per mg of tissue that would allow for a volume suitable for conducting the tests required. A more concentrated sample would have shown a higher IgA titre; however, it is likely the trend between groups would have remained the same.
Washes were already started at a high dilution and titrated down as the reviewer suggested, the resulting values are the titration where the detection passes the average blank + 3 SD, allowing for very low detection of IgA. Testing of mucosal secretions at other sites is being explored in our further studies but was not possible in the confines of the setup of these experiments. We mocked up the graphs with a Log2 scale but did not feel it made the data clearer so have retained a Log10 scale. We have amended the colour scheme as in Figure 1 and hope this is satisfactory to the reviewer.
Figure 3A and B: What GAS strain was used for this study (emm type, animal passaged/not passaged, origin of the strain)? Figure 3A did the authors assess surface binding with mucosal samples? Have they assessed mucosal IgG responses? In Figure 3A, why didn’t they assess GAS binding with Alum adjuvanted/non adjuvanted samples? For Figure 3B, did they dilute the sera to determine non-specific killing, OPK titre and opsonic index? Please refer to manuscript: DOI: 10.1007/978-1-0716-0467-0_26.
M1 strain NCTC8198 was used in this study as the only M-typed strain available for purchase at the time from NCTC. This information has been added to the manuscript. We did assess the surface binding with mucosal samples but this wasn’t optimised within the confines of the study and only showed a minor shift. We did not test the mucosal IgG response as this would be unlikely to add value to the data as IgG is transduced from circulation into mucosal tissues. Apologies, Figure 3A was mislabelled and should have read SC group with/without Alum. This has been corrected. We have found with the OPA that dilution of sera removes killing activity so have only been able to compare killing activity when neat undiluted sera are used and are unable to determine OPK titre and opsonic indices. Though the ability to perform this has been published, anecdotally most laboratories have difficulty with this aspect of the HL60 based assay, particularly with mouse sera.
Lines 551-555: There is mention of AS04 adjuvant without any mention of the role of T-cells/Th1 in GAS vaccine design. Also, worth mentioning AS01 adjuvant as this is a liposomal formulation and has the potential to be administered IN.
We have included AS01 adjuvant, however this was not intended to be an exhaustive analysis of all the adjuvant systems available for mucosal delivery. We have also added into the text the importance of selecting the appropriate adjuvant for the required immune response. As there are still no correlates of protection with GAS it is worth noting that the role of adjuvanted responses has not been investigated in protection generated. We’re hopeful the coming years will answer some of these questions.
Line 555: Very risky to mention potential whole-cell vaccine design for GAS. Please consider removing or elaborating with a rationale on this.
Apologies, we weren’t referring to whole cells of GAS, but rather whole cell vectors (such as L. lactis) or cell-like structures (e.g. VLPs). We have clarified the text.
Line 556: Recent studies with CAF01 show that priming with Ag+CAF01 (SC/IM) followed by pulling with Ag alone (no adjuvant) also induces protective Ab responses in the mucosa and protection. Please refer to publication: DOI: 10.1128/mBio.03537-20.
This has been added to the text.
Reviewer 3 Report
Comments and Suggestions for Authors
The manuscript submitted by Shaw et al. reported the comparison the immune response elicited by GAS protein antigens via three different immunization manner. They discovered that mucosal immunization with adjuvant could induce both systemic IgG and mucosal IgA antibodies, and these antibodies could show bactericidal activities and block GAS binding to HaCaT cells. Further study also indicated that the corresponding immune sera could neutralized SpyCEP activity to IL-8 and Mac/IdeS IgG cleavage activity. This study demonstrated mucosal immunization is a viable alternative manner for GAS vaccine immunization. The manuscript is well organized and executed, thus it is recommended to be publishable after several minor revision.
-- Page 3, line 110, "immunizsation" should be "immunization";
--Page 3, line 127, "... to incubation at 4oC 3 h with ..." should be "... to incubation at 4oC for 3 h with ..."; Page 4, line 181, "... grown at 37oC 5% CO2 for 3 days ..." should be "... grown at 37oC with 5% CO2 for 3 days ...";
--Page 6, line 280, "PBS control (IN)" should be "PBS control”; the same error at Page 7, line303;
--Page 7, subheading "3.2.1 IgA response at D21 and D36" should be "3.2.2 IgA response at D21 and D36".
Author Response
Thank you for taking the time to review our manuscript, we have amended the text as recommended and hope this is satisfactory to the reviewer.
Reviewer 4 Report
Comments and Suggestions for Authors
Shaw et al address Group A Streptococci (GAS) as an important medical problem by vaccination. GAS may colonize mucosal surfaces but may also spread systemically.
The authors use the recombinant antigens SpyCEP, Cpa, Mac and MalE to induce a broad humoral immune response in BALB/c mice, the preferred mouse strain for humoral immune responses.
The humoral immune response was broadly analyzed by determining the serum and mucosal IgG and IgA titer as well as the potency to kill GAS infected cells.
1: 40 micrograms of protein was used for all immunizations. For the nasal immunization an additional 10 micrograms of cholera toxin was added in a total volume of 10 microl. The reviewer assumes that this large amount of protein is partly swallowed and/or reached the lung. Was the spread of the antigen in various organs analyzed? This may be important for a possible translation in humans where the dose/weight ratio of a vaccine and thus the location of the initiation of an immune response is likely different, if oral/nasal immunization is proposed for humans too.
2: Only IgG and IgA was analyzed. At least IgG must be T cell dependent. There is no data on T cells or isotypes of the IgGs to define a possible Th1/2 bias.
3: There is no data on the memory response. This might be important for IgA and possibly the location of the immune onset as IgA can be induced T cell dependent and is likely longer lasting than the IgA T cell independent immune response.
4: In the past we have seen wonderful serological immune responses after vaccination with GAS antigens with little or even devastating immune responses after challenge. There are no data on the efficacy of the proposed vaccine. So what is the value of the information?
Author Response
Thank you for taking the time to review our manuscript, we appreciate your comments particularly on memory response and this will be a focus of our future studies. We hope our responses to comments and amendments to the manuscript are satisfactory to the reviewer. Specific responses are in italics below.
1: 40 micrograms of protein was used for all immunizations. For the nasal immunization an additional 10 micrograms of cholera toxin was added in a total volume of 10 microl. The reviewer assumes that this large amount of protein is partly swallowed and/or reached the lung. Was the spread of the antigen in various organs analyzed? This may be important for a possible translation in humans where the dose/weight ratio of a vaccine and thus the location of the initiation of an immune response is likely different, if oral/nasal immunization is proposed for humans too.
The sample volumes used (20 ul SL and 10 ul IN) and the method of immunization (see clarified M&M) were such that there is minimal risk of swallowing or inhaling the preparations. We have elaborated on the method of immunization to explain this. Studying the spread of the antigen was not in the scope of the current project so was not addressed.
2: Only IgG and IgA was analyzed. At least IgG must be T cell dependent. There is no data on T cells or isotypes of the IgGs to define a possible Th1/2 bias.
Analysis of IgG subclasses or T-cell response was not in the scope of this study and will be included in future investigations. We have added into the text the importance of future studies focusing on a T cell response.
3: There is no data on the memory response. This might be important for IgA and possibly the location of the immune onset as IgA can be induced T cell dependent and is likely longer lasting than the IgA T cell independent immune response.
We realize the importance of looking at the memory response, which was not in the scope of the current project but will be a focus of further studies.
4: In the past we have seen wonderful serological immune responses after vaccination with GAS antigens with little or even devastating immune responses after challenge. There are no data on the efficacy of the proposed vaccine. So what is the value of the information?
This was an initial study to assess and compare the immune response and in vitro functionality between different routes of immunization. The aim was to determine whether mucosal immunization could compete with subcutaneous immunization. When this serological and mucosal immune response has been optimized, we aim to investigate the protective effect in animal challenge models in future projects, but this was not within the scope of the initial study. The value of the information was to demonstrate that mucosal immunization is a viable alternative to subcutaneous immunization and could be superior as it can induce systemic and mucosal functionally active antibodies, as demonstrated in our study using serum and mucosal secretions.
Round 2
Reviewer 2 Report
Comments and Suggestions for Authors
Authors have satisfactorily addressed all comments.
Author Response
Thank you, we appreciate the feedback you gave and are glad you are satisfied with our responses and changes.
Reviewer 4 Report
Comments and Suggestions for Authors
I still think it is an incomplete story.
Author Response
Thank you, we appreciate your view and as such with recommendation of the editor are changing the paper to be a pilot study. Many of your recommendations are in process for our next study where we are trying to optimise the response and some of your suggestions will enable us to make it a more intensive survey of the immune response and protection. Thank you again for your review.